# Enhanced mitophagy in bronchial fibroblasts from severe asthmatic patients

**Rakhee K. Ramakrishnan**[1,2], **Khuloud Bajbouj**[1,2], **Mahmood Y. Hachim**[3], **Andrea K. Mogas**[4], **Bassam Mahboub**[1,5], **Ronald Olivenstein**[4], **Rifat Hamoudi**[1,2], **Rabih Halwani**[1,2], **Qutayba Hamid**[1,4]*

1 College of Medicine, University of Sharjah, Sharjah, United Arab Emirates, 2 Sharjah Institute for Medical Research, University of Sharjah, Sharjah, United Arab Emirates, 3 College of Medicine, Mohammed Bin Rashid University, Dubai, United Arab Emirates, 4 Meakins-Christie Laboratories, McGill University, Montreal, QC, Canada, 5 Rashid Hospital, Dubai Health Authority, Dubai, United Arab Emirates

* qalheialy@sharjah.ac.ae

## Abstract

### Background

Sub-epithelial fibrosis is a characteristic feature of airway remodeling in asthma which correlates with disease severity. Current asthma medications are ineffective in treating fibrosis. In this study, we aimed to investigate the mitochondrial phenotype in fibroblasts isolated from airway biopsies of non-asthmatic and severe asthmatic subjects by examining mitophagy as a mechanism contributing to fibroblast persistence and thereby, fibrosis in severe asthma.

### Methods

Bioinformatics analysis of publicly available transcriptomic data was performed to identify the top enriched pathways in asthmatic fibroblasts. Endogenous expression of mitophagy markers in severe asthmatic and non-asthmatic fibroblasts was determined using qRT-PCR, western blot and immunofluorescence. Mitophagy flux was examined by using lysosomal protease inhibitors, E64d and pepstatin A. Mitochondrial membrane potential and metabolic activity were also evaluated using JC-1 assay and MTT assay, respectively.

### Results

Bioinformatics analysis revealed the enrichment of Pink/Parkin-mediated mitophagy in asthmatic fibroblasts compared to healthy controls. In severe asthmatic fibroblasts, the differential expression of mitophagy genes, PINK1 and PRKN, was accompanied by the accumulation of PINK1, Parkin and other mitophagy proteins at baseline. The further accumulation of endogenous LC3BII, p62 and PINK1 in the presence of E64d and pepstatin A in severe asthmatic fibroblasts reinforced their enhanced mitophagy flux. Significantly reduced mitochondrial membrane potential and metabolic activity were also demonstrated at baseline confirming the impairment in mitochondrial function in severe asthmatic fibroblasts. Interestingly, these fibroblasts displayed neither an apoptotic nor senescent phenotype but a pro-fibrotic phenotype with an adaptive survival mechanism triggered by increased AMPKα phosphorylation and mitochondrial biogenesis.

**Data Availability Statement:** All relevant data are within the manuscript and its Supporting Information files.

**Funding:** This study was supported by a collaborative grant (16010902009-P) from the University of Sharjah. The funders had no role in study design, data collection and analysis, decision to publish, or preparation of the manuscript.

**Competing interests:** The authors have declared that no competing interests exist.

**Abbreviations:** ECM, Extracellular matrix; AEC, Alveolar epithelial cells; QC, Quality control; PINK1, PTEN-induced putative kinase 1; COPD, Chronic obstructive pulmonary disease; IPF, Idiopathic pulmonary fibrosis; DMEM, Dulbecco's Modified Eagle's Medium; qRT-PCR, Quantitative Real Time-Polymerase Chain Reaction; FN1, Fibronectin; ATG5, Autophagy related 5; LC3B, Microtubule-associated protein 1 light chain 3B; SQSTM1/p62, Sequestosome-1; LAMP2, Lysosomal-associated membrane protein 2; PRKN, Parkin; ΔΨm, Mitochondrial membrane potential; AMPK, AMP-activated protein kinase; SIRT1, Sirtuin 1; PGC1α, Peroxisome proliferator-activated receptor gamma coactivator 1 alpha.

## Conclusions

Our results demonstrated a role for mitophagy in the pathogenesis of severe asthma where the enhanced turnover of damaged mitochondria may contribute to fibrosis in severe asthma by promoting the persistence and pro-fibrotic phenotype of fibroblasts.

## Introduction

Sub-epithelial fibrosis is a characteristic feature of airway remodeling documented across all degrees of asthma severity [1, 2]. In addition to being the prime source of extracellular matrix (ECM) proteins [3], fibroblasts are important in maintaining a state of chronic pulmonary inflammation through the production of a variety of cytokines, chemokines, proteases and lipid mediators [4]. Current asthma therapy, including inhaled corticosteroids, leukotriene antagonists and long-acting beta-agonists, have shown limited effectiveness in arresting or reversing sub-epithelial fibrosis [5], necessitating the identification of novel fibrotic mechanisms.

Mitochondrial dysfunction is suggested to have downstream consequences on key aspects of asthma pathophysiology, including fibrosis, proliferation, apoptosis, response to oxidative stress, calcium regulation and airway contractility [6]. Mitochondrial dysfunction in the different airway cell populations, including alveolar epithelial cells (AECs), fibroblasts and immune cells, can contribute towards the fibrotic process [7], by stimulating AEC-derived cytokines leading to activation of myofibroblasts [8].

Mitochondrial homeostasis is ensured by the coordinated operations of quality control (QC) mechanisms including mitophagy and mitochondrial biogenesis. PTEN-induced putative kinase 1 (PINK1) and E3 ubiquitin ligase Parkin are well known regulators of mitophagy. Their coordinated activities are capable of sensing and triggering the removal of damaged mitochondria [9]. While the defective mitochondria are selectively recycled via mitophagy, new functional mitochondria are replenished into the pool through biogenesis in order to maintain a functional mitochondrial network under stressful environments.

Mitophagy is usually enhanced as an early protective response to promote cell survival. Emerging evidence suggests that autophagy and mitophagy can play both protective as well as detrimental roles in human pulmonary diseases albeit in a cell type-specific manner [10, 11]. Disruptions in mitochondrial QC have been extensively studied in the pathogenesis of other chronic lung diseases, including chronic obstructive pulmonary disease (COPD) and idiopathic pulmonary fibrosis (IPF) [12, 13], but comparatively less explored in asthma.

In fact, oxidative stress and mitochondrial dysfunction have been implicated in the development and progression of asthma [14, 15]. Microscopic analysis of bronchial epithelium from asthmatic children revealed abnormal ultrastructural changes in the mitochondria providing one of the early evidences of mitochondrial abnormality in asthma [16]. Mechanistically, oxidative damage-induced mitochondrial dysfunction drives the molecular processes responsible for the development of allergic airway inflammation implying that mitochondrial defects could pose as risk factors for severe allergic disorders in atopic individuals [15].

Previous work by our group had demonstrated the involvement of autophagy in asthma pathogenesis. Increased number of autophagosomes were detected in fibroblasts within bronchial biopsy tissue from a moderately severe asthmatic [17]. We also showed a positive correlation between *ATG5* gene expression and collagen deposition in the airways of refractory moderate-to-severe asthmatics suggesting that dysregulation of autophagy may promote sub-epithelial fibrosis in severe asthmatic airways [18]. The aim of this study was to analyze the

mitochondrial QC mechanism of mitophagy in bronchial fibroblasts from severe asthmatic (S-As) patients. These fibroblasts demonstrated increased mitophagy flux, however, with enhanced mitochondrial turnover. This may contribute to increased persistence of these fibroblasts despite their mitochondrial damage and hence, the aberrant pro-fibrotic phenotype observed in severe asthma.

## Materials and methods

### Bioinformatics analysis

In order to identify the top enriched pathways in asthmatic fibroblasts compared to their healthy counterparts, we explored publicly available gene expression databases. We searched datasets where pulmonary fibroblasts were compared between asthmatic and healthy controls. One of the few studies that fulfilled the criteria was GSE27335 [19], where both parenchymal and bronchial fibroblasts were taken into consideration. Microarray analysis was performed on 12 different matched pairs of fibroblasts (4 pairs from normal subjects and 8 pairs from asthmatics). The normalized gene expression of all the samples was extracted and subjected to differential expression analysis to identify the top upregulated and downregulated genes in asthmatic fibroblasts compared to healthy controls. Differential gene expression analysis and Gene Ontology enrichment analysis using reactome database were performed using the BioJupies tool [20].

### Fibroblast cell culture

Human primary bronchial fibroblasts derived from endobronchial tissue specimens were obtained from the Quebec Respiratory Health Research Network Tissue Bank (McGill University Health Centre (MUHC)/ Meakins-Christie Laboratories Tissue Bank, Montreal, Canada), as described previously [21]. The original study was approved by the MUHC Research Ethics Board (2003–1879) and the subjects had provided written informed consent. Fibroblasts used in this study were age-matched and chosen from subjects who were non-smokers. The mean age of the subjects was 43.7 ± 12.5 years for the healthy controls and 43.3 ± 8.3 years for the severe asthmatics. Bronchial fibroblasts isolated from three severe asthmatic and three non-asthmatic healthy control subjects were maintained in Dulbecco's Modified Eagle's Medium (DMEM) supplemented with 10% Fetal Bovine Serum (FBS) and 1% Penicillin-Streptomycin in a humidified 5% $CO_2$/37˚C incubator. For experiments, the cells were seeded into 12- or 96-well plates at a cellular density of $5x10^4$ and $2x10^3$ cells/well, respectively. At ~70% confluency, they were serum-starved in FBS-free DMEM complete medium for 24 hours, and cultured in DMEM complete medium thereafter for the specified amount of time. The primary fibroblasts included in this study exhibited doubling times varying from 30–48 hours. Therefore, serum starvation was performed prior to all experiments in order to synchronize the population of proliferating cells and to provide a more reproducible experimental condition [22]. In order to investigate the lysosomal turnover of mitophagy markers, the fibroblasts were co-incubated with 10μg/ml of E64d (Santa Cruz, Cat. No. sc-201280A) and 10μg/ml of Pepstatin A (Santa Cruz, Cat. No. sc-45036). In order to inhibit autophagy, the cells were treated with 1mM 3-methyl adenine (3-MA) (R&D Systems, Cat. No. 3977) for 48 hours. The cells were used within passage 8 in the experiments described below. All cell culture reagents were purchased from Sigma-Aldrich.

### Quantitative real time-polymerase chain reaction (qRT-PCR)

Total RNA was extracted from cell pellets using the RNeasy Mini Kit (Qiagen, Cat. No. 74106). RNA quality and concentrations were determined by Nanodrop (Thermo Scientific)

**Table 1. Sequence list of primers used for qRT-PCR.**

| Genes | Forward Primer Sequence (5'-3') | Reverse Primer Sequence (5'-3') |
|---|---|---|
| COL1A1 | GATTGACCCCAACCAAGGCTG | GCCGAACCAGACATGCCTC |
| COL3A1 | GATCAGGCCAGTGGAAATG | GTGTGTTTCGTGCAACCATC |
| COL5A1 | GTCGATCCTAACCAAGGATGC | GAACCAGGAGCCCGGGTTTTC |
| FN1 | CTGGGAACACTTACCGAGTGGG | CCACCAGTCTCATGTGGTCTCC |
| IL6 | GAAAGCAGCAAAGAGGCAC | GCACAGCTCTGGCTTGTTCC |
| IL11 | GTGGCCAGATACAGCTGTCGC | GGTAGGACAGTAGGTCCGCTC |
| IL8 | CCACACTGCGCCAACACAG | CTTCTCCACAACCCTCTGC |
| GROα | CTGCAGGGAATTCACCCCAAG | GATGCAGGATTGAGGCAAGC |
| ATG5 | GACCAGTTTTGGGCCATCAATC | GTGCAACTGTCCATCTGCAGC |
| LC3B | GAACGGACACAGCATGGTCAGC | ACGTCTCCTGGGAGGCATAG |
| p62 | TTGTACCCACATCTCCCGCCA | TACTGGATGGTGTCCAGAGCCG |
| LAMP2 | AACTTCAACAGTGGCACCCACC | AGTGATGTTCAGCTGCAGCCCC |
| PINK1 | CCTGCGCCAGTACCTTTGTGT | TGGGTCCAGCTCCACAAGGATG |
| PRKN | CTCCAGCCATGGTTTCCCAGTG | CCAGGTCACAATTCTGCACAGTC |
| 18s | TGACTCAACACGGGAAACC | TCGCTCCACCAACTAAGAAC |

spectrophotometric measurements. cDNA synthesis was performed from 300ng of RNA using the FIRESCript RT cDNA Synthesis Kit (Solis Biodyne, Cat. No. 06-15-00050) in the Veriti Thermal Cycler (Applied Biosystems). qRT-PCR reactions were set up using the 5x Hot Fire-Pol EvaGreen qRT-PCR SuperMix (Solis Biodyne, Cat. No. 08-36-00001) in QuantStudio 3 Real-Time PCR System (Applied Biosystems). The primers used are listed in Table 1. Gene expression was analyzed using the Comparative $C_T$ ($\Delta\Delta C_T$) method after normalization to the housekeeping gene 18s rRNA. All results are presented as fold expression change compared to non-asthmatic healthy controls.

## Western blot

The cell pellets were lysed in RIPA lysis buffer (50mM Tris, 150mM NaCl, 1% sodium deoxycholate, 0.1% sodium-dodecyl-sulphate (SDS), 1% Triton X-100, pH7.5) supplemented with 1x Protease Inhibitor Cocktail (Sigma-Aldrich, Cat. No. P2714) and 1mM phenylmethylsulfonyl fluoride (Sigma-Aldrich, Cat. No. P7626). The protein lysates were quantified using the Protein Assay Kit II (Bio-Rad, Cat. No. 5000002). After routine electrophoresis and blotting steps, the primary and secondary antibody-probed blots were developed using the Clarity Western ECL Substrate (Bio-Rad, Cat. No. 170–5060) in the ChemiDoc Touch Gel and Western Blot Imaging System (Bio-Rad). The following antibodies were used in this study–anti-LC3B (abcam, Cat. No. ab51520, 1:5000 dilution), Mitophagy Antibody Sampler Kit (Cell Signaling Technology, Cat. No. 43110T, 1:1000 dilution), anti-LAMP2A (abcam, Cat. No. ab125068, 1:2000 dilution), phospho-AMPKα (Thr172) (Cell Signaling Technology, Cat. No. 2535T, 1:1000 dilution), anti-AMPKα2 (abcam, Cat. No. ab3760, 1:1000 dilution), anti-SIRT1 (Cell Signaling Technology, Cat. No. 8469s, 1:2000 dilution) and anti-PGC1α (Novus Biologicals, Cat. No. NBP1-04676, 1:2000 dilution). Anti-β-actin (Sigma-Aldrich, Cat. No. A5441, 1:10000 dilution) was used as the loading control. Anti-rabbit IgG, HRP-linked (Cell Signaling Technology, Cat. No. 7074S, 1:3000 dilution) and anti-mouse IgG, HRP-linked (Cell Signaling Technology, Cat. No. 7076S, 1:3000 dilution) secondary antibodies were used. Image Lab software (Bio-Rad) was used to detect and quantify the protein bands.

## Autophagy assay

Autophagosomal levels were assessed in the bronchial fibroblasts using the Autophagy Assay Kit (Sigma-Aldrich, Cat. No. MAK138). Autophagy detection was performed as per kit instructions. Briefly, the fibroblasts were stained with the Autophagosome Detection Reagent for 30 minutes at 37˚C in the dark. The fluorescence intensity was measured at an excitation of 360nm and emission of 520nm in Synergy HTX fluorescence reader (BioTek) and imaged using Olympus BX51 fluorescence microscope.

## Immunofluorescence assays

Mitophagy tracking was performed using fibroblasts transduced with Premo Autophagy Sensor LC3B-GFP (Invitrogen, Cat. No. P36235) and stained thereafter for PINK1. Cells were transduced with LC3B-GFP as per manufacturer instructions, cultured for 24 hours and stained with anti-PINK1 (Novus Biologicals, Cat. No. BC100-494, 1:500 dilution) primary antibody and PE-conjugated secondary antibody (Molecular Probes, Cat. No. P-2771MP, 1:500 dilution).

Mitochondria and lysosomes were tracked by staining fibroblasts using MitoTracker Green (Invitrogen, Cat. No. M7514) and LysoTracker Deep Red (Invitrogen, Cat. No. L12492) fluorescent probes, respectively. Cells were stained with 200nM Mitotracker Green and 100nM Lyostracker Deep Red for 15 min at 37˚C before imaging. Images were taken by the Nikon Eclipse Ti confocal microscope.

## Mitochondrial membrane potential assay

Mitochondrial membrane potential was measured using the cationic dye tetraethylbenzimidazolylcarbocyanine iodide (JC-1; abcam, Cat. No. ab113850) that aggregates in energized mitochondria, in accordance with manufacturer's instructions. Briefly, the fibroblasts were stained with 10μM JC-1 solution for 10 minutes at 37˚C in the dark. The microplate was read in the Varioskan Flash multi-mode plate reader (Thermo Scientific) at an excitation wavelength of 475nm and emission wavelength of 590nm to measure the JC-1 aggregate signal, which is an indicator of energized mitochondria.

## Mitochondrial metabolic activity assay

After the desired incubation period, the cells were incubated with MTT (3-(4,5-dimethylthiazol-2-yl)-2,5-diphenyltetrazolium bromide) solution (5mg/ml in PBS; Sigma-Aldrich, Cat. No. M5655) for 3 hours at 37˚C and dimethylsulfoxide (DMSO; Sigma-Aldrich, Cat. No. D8418) was used to solubilize the formazan crystals. The absorbance was read at 570nm in Synergy H1 multi-mode microplate reader (BioTek).

## Annexin V apoptosis assay

Apoptosis was measured in fibroblasts using the PE Annexin V Apoptosis Detection Kit with 7-AAD (Biolegend, Cat. No. 640934), according to manufacturer instructions. Briefly, the cells were labelled with Annexin V-PE and 7-AAD for 20 minutes at room temperature in the dark. The cells were then analyzed using the BD FACSAria III flow cytometer and the acquired data analyzed using FlowJo v10 software.

## β-galactosidase senescence assay

Cellular senescence was measured in fibroblasts using the β-galactosidase staining kit (abcam, Cat. No. ab102534) as per kit protocol. In brief, the cells were fixed and incubated in X-gal

staining solution overnight at 37°C inside a sealable bag. Images were taken using a bright-field microscope (Olympus BX51).

## Statistical analysis

All results are presented as mean ± standard error of the mean (SEM) from 2–4 independent experiments using GraphPad Prism 6.0 software. Statistical comparisons between the groups (severe asthmatic and non-asthmatic control fibroblasts) were performed using unpaired Student t-test. Unpaired t-test with multiple comparisons using the Holm-Sidak method was used to ascertain statistical significance in the presence of treatments. A $p$ value $< 0.05$ was considered statistically significant.

## Results

### 1. Mitophagy related pathways are among the top enriched pathways in asthmatic fibroblasts

To investigate the differential regulation of genes related to autophagy and mitophagy in asthmatic fibroblasts, we explored the publicly available gene expression database GSE27335; which compared fibroblasts from asthmatics versus healthy controls. Differential gene expression analysis (Fig 1A and 1B) followed by Gene Ontology enrichment analysis using reactome database (Fig 1C) indicated that Pink/Parkin Mediated Mitophagy pathway was among the

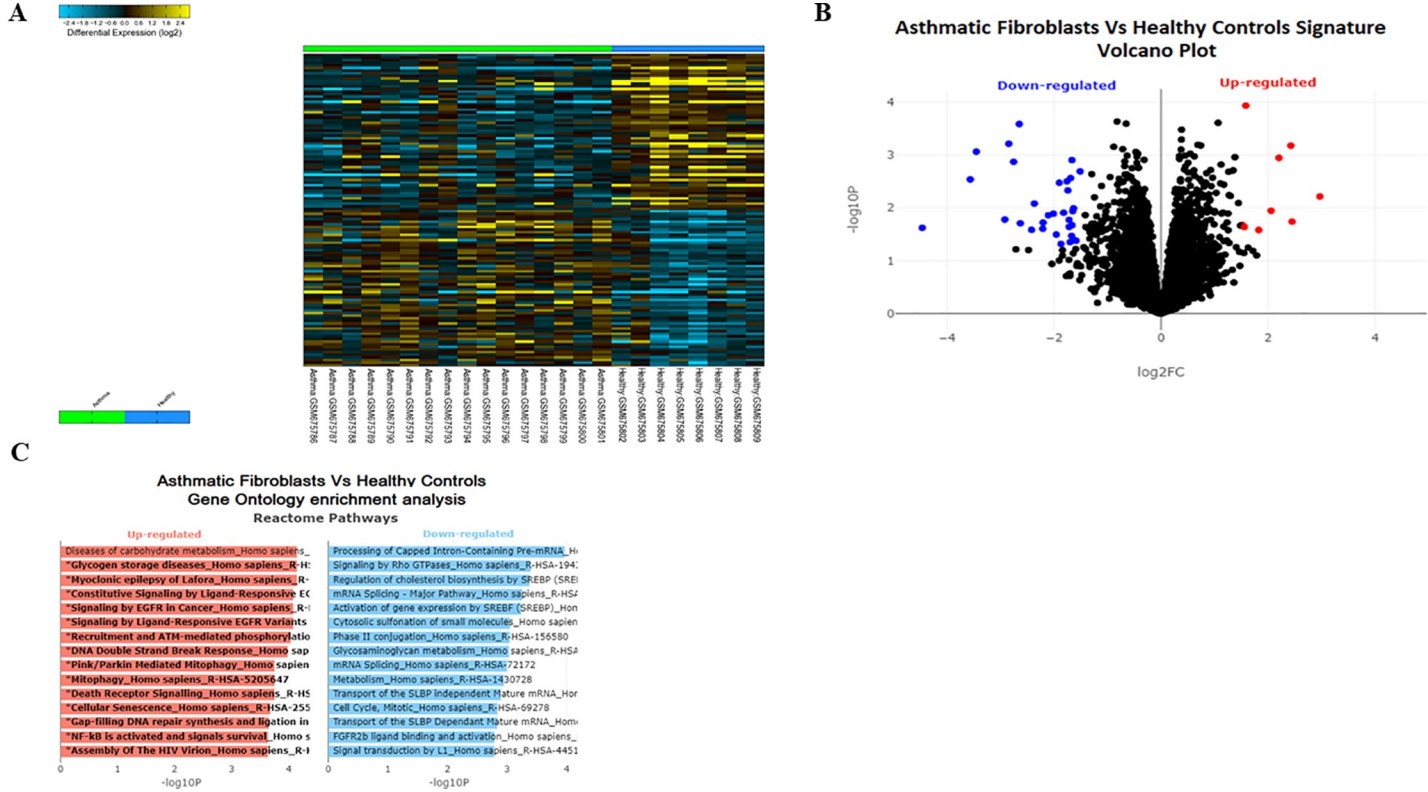

**Fig 1. PINK/Parkin mediated mitophagy is among the top enriched pathways in asthmatic fibroblasts. (A)** Heatmap of differentially expressed genes (DEG) between asthmatic and healthy fibroblasts. **(B)** The log2-fold change and statistical significance of each gene was calculated by performing differential gene expression analysis. Each point in the plot represents a gene. Red points indicate significantly up-regulated genes and blue points indicate significantly down-regulated genes (logFC threshold = 1.5 and p-value threshold = 0.05). **(C)** Gene Ontology enrichment analysis was generated using Enrichr tool. The x-axis indicates the -log10(p-value) for each term. Significant terms are highlighted in bold. Only pathways with False Discovery Rate (FDR) less than 0.05 were selected.

top upregulated pathways in asthmatic fibroblasts when compared to healthy. This implicated the dysregulation of mitophagy in the pathogenesis of asthma and we decided to explore further the mechanism of this dysregulation and its contribution to fibrosis.

## 2. Increase in basal autophagy in severe asthmatic fibroblasts

Previously, we reported the dysregulation of autophagy in asthmatic airway tissues [17, 18]. Therefore, we first examined the basal autophagy levels in severe asthmatic (S-As) and control fibroblasts. The primary fibroblasts used in this study were previously described [21]. A significant increase in autophagic vacuoles was observed in the S-As fibroblasts compared to the control fibroblasts (Fig 2A). A 1.35-fold increase in autophagosomal fluorescence intensity was detected in S-As fibroblasts in comparison to control (p = 0.0314), suggesting an increase in the number of autophagosomes at baseline in S-As fibroblasts.

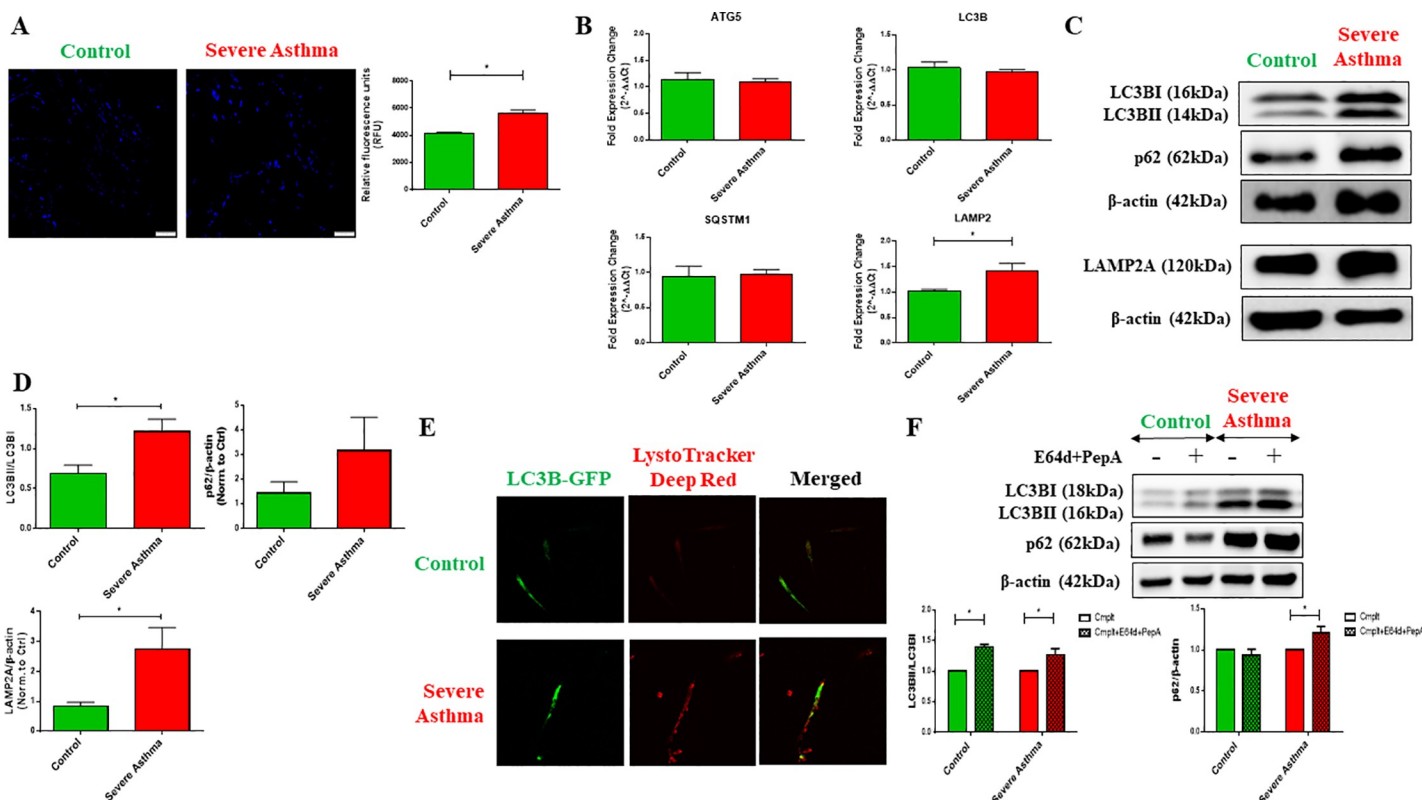

**Fig 2. Increase in basal autophagy in severe asthmatic fibroblasts.** The control and severe asthmatic (S-As) fibroblasts were cultured in DMEM complete medium for 4 hours post serum-starvation to measure autophagy at baseline. **(A)** To measure autophagosomal levels, the fibroblasts were stained with Autophagosome Detection Reagent for 30 minutes at 37°C in the dark and fluorescent readings were taken using a fluorescence microscope and plate reader. Representative images showing fluorescent staining of autophagosomal vacuoles (blue) in control and S-As fibroblasts (left panel). Quantitative representation of autophagosomal levels in relative fluorescence units (RFU) (right panel). **(B)** Under basal conditions, mRNA expression of autophagy markers, *ATG5*, *LC3B*, *SQSTM1/p62* and *LAMP2*, in control and S-As fibroblasts was analysed by qRT-PCR and expressed as fold expression change relative to control fibroblasts post normalization to housekeeping gene 18s rRNA. **(C)** Representative immunoblots depicting protein levels of LC3B, p62 and LAMP2A in control and S-As fibroblasts. β-actin was used as loading control. **(D)** Densitometric analysis of LC3B lipidation represented as the ratio of LC3BII to LC3BI, p62 and LAMP2A levels in control and S-As fibroblasts. **(E)** The fibroblasts were cultured in DMEM complete medium for 48 hours post serum-starvation for immunofluorescence measurements at baseline. Representative images depicting fluorescent staining of autophagosomes using LC3B-GFP (green) and lysosomes using LysoTracker Deep Red (red). **(F)** The fibroblasts were cultured in the presence of lysosomal protease inhibitors, E64d and pepstatin A, for 6 hours post serum-starvation. Cell lysates were subjected to immunoblot analysis of autophagy proteins LC3B and p62. β-actin was used as loading control (top panel). Densitometric analysis of LC3B lipidation represented as the ratio of LC3BII to LC3BI and p62 levels in control and S-As fibroblasts upon treatment with E64d and pepstatin A (bottom panel). Graphical data are represented as mean ± SEM from 2–4 independent experiments with at least 3 unique donors in each group. *p < 0.05, determined by unpaired two-tailed Student t-test (Control vs Severe Asthma) or unpaired t-test with multiple comparisons using the Holm-Sidak method (Cmplt vs Cmplt+E64d+PepA).

The gene expression levels of autophagy markers, autophagy related 5 (ATG5), microtubule-associated protein 1 light chain 3B (LC3B), sequestosome-1 (SQSTM1/p62) and lysosomal-associated membrane protein 2 (LAMP2), were then determined. The autophagy related genes, *ATG5*, *LC3B* and *SQSTM1*, did not appear to be differentially expressed between the S-As and control fibroblasts (Fig 2B). However, we detected an upregulation in the gene expression of *LAMP2* (p = 0.0357) (Fig 2B), that encodes a lysosomal protein responsible for lysosomal stability and lysosomal degradation of autophagic vacuoles [23, 24].

We next performed western blot analysis to determine the levels of autophagy proteins at baseline. Upon activation of autophagy, the cytosolic LC3BI is converted to the autophagosomal membrane-bound LC3BII, which serves as an early indicator of autophagosomal formation. The LC3BII levels as well as LC3B lipidation (conversion of LC3BI to LC3BII represented as LC3BII/LC3BI ratio) were increased in S-As fibroblasts compared to control (p = 0.0234) (Fig 2C and 2D). Additionally, the p62 levels also showed an increased trend, but it did not reach statistical significance (Fig 2C and 2D).

LC3BII and p62 accumulation can signify either enhanced autophagy initiation or defective clearance of autophagic vacuoles by lysosomes. To exclude lysosomal defect, we determined the LAMP2A protein levels using western blot analysis and used the LysoTracker probe that stains lysosomes. LAMP2A was significantly increased in S-As fibroblasts (p = 0.0304) (Fig 2C and 2D) in agreement with the increased *LAMP2* gene expression (Fig 2B). Enhanced LysoTracker fluorescence was also detected in these fibroblasts (Fig 2E), suggesting increased lysosomal activity and thus, excluding the possibility of defective autophagy clearance. We further, transduced the fibroblasts with LC3B-GFP to assess the co-localization of autophagosomes and lysosomes. Increased co-localization of LC3B and LysoTracker probes was indicated in S-As fibroblasts when compared to control fibroblasts (Fig 2E), suggesting increased autophagy flux in S-As fibroblasts.

Autophagy being a dynamic process requires the use of multiple assays to monitor the entire process as well as to appropriately interpret the results [25]. The lysosomal turnover of LC3BII and p62 reportedly provides a more accurate indication of autophagic activity than its baseline levels [26]. We, therefore, cultured the fibroblasts in the presence of two lysosomal protease inhibitors, E64d and pepstatin A, for a period of 6 hours to monitor the kinetics of autophagy turnover. LC3BII was the predominant form in S-As fibroblasts when compared to control (Fig 2F). In the presence of E64d and pepstatin A, accumulation of LC3BII was observed in both group of fibroblasts (Fig 2F). p62 also showed further accumulation upon treatment with E64d and pepstatin A in S-As fibroblasts (Fig 2F), indicating their lysosomal turnover through autophagy. The above findings suggest an increase in basal autophagosome formation as well as autophagy flux in S-As fibroblasts when compared to their healthy counterparts.

## 3. Accumulation of endogenous full-length PINK1 in severe asthmatic fibroblasts

We next aimed to determine if the increased autophagy exhibited by S-As fibroblasts was linked to their mitochondrial phenotype as the mitochondrial QC mechanism of mitophagy uses the autophagy machinery for completion. Since the bioinformatics analysis indicated enrichment of Pink/Parkin-mediated mitophagy genes in asthmatic fibroblasts, we first examined the gene expression of mitophagy markers, *PINK1* and *PRKN*, by qRT-PCR. The mRNA expression of *PINK1* (p = 0.0820) and *PRKN* (p = 0.0055) was upregulated in S-As fibroblasts when compared to control (Fig 3A). The differential expression of *PINK1* and *PRKN* genes thus, supported the bioinformatics analysis and indicated the activation of Pink/Parkin-mediated mitophagy pathway in S-As fibroblasts.

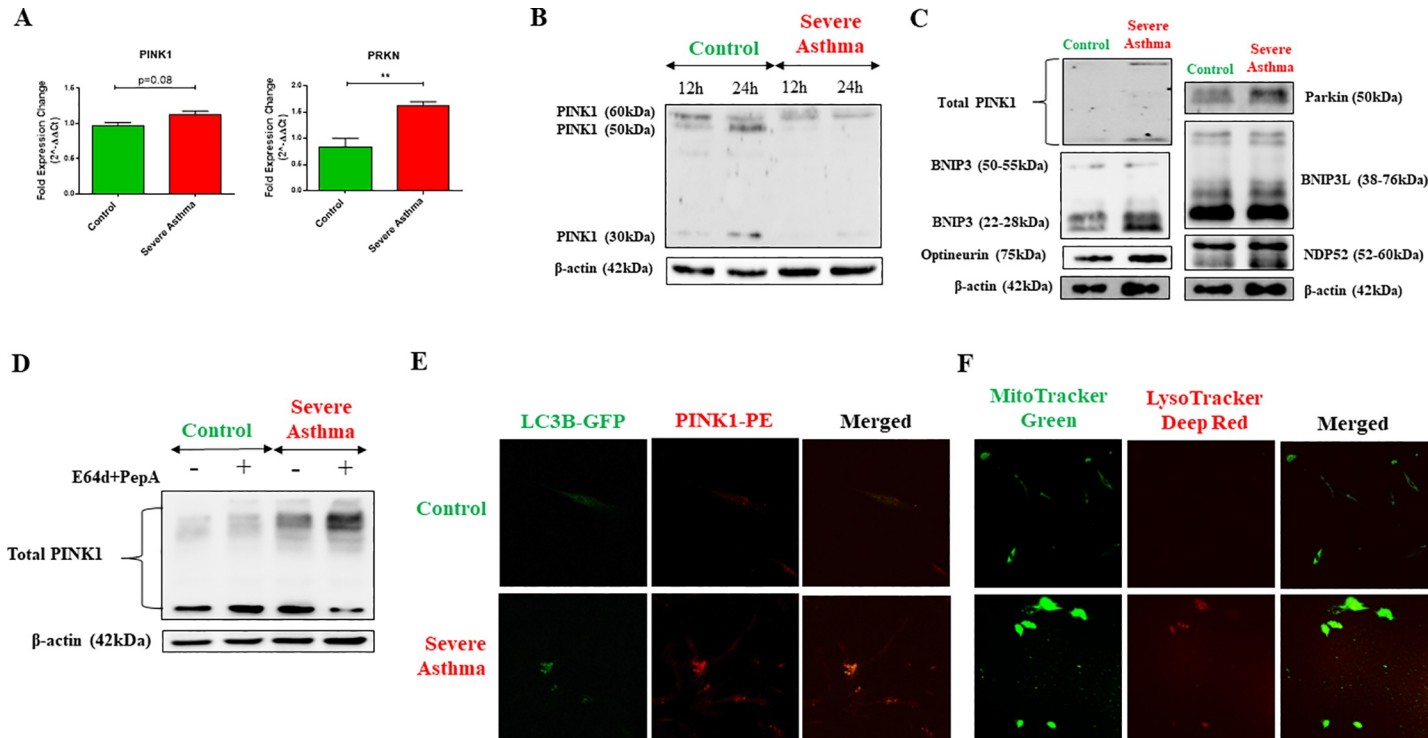

**Fig 3. Stabilization of PINK1 in severe asthmatic fibroblasts.** (A) The fibroblasts were cultured in DMEM complete medium for 4 hours post serum-starvation. Under basal conditions, mRNA expression of mitophagy markers, *PINK1* and *PRKN*, in control and S-As fibroblasts was analysed by qRT-PCR and expressed as fold expression change relative to control fibroblasts post normalization to housekeeping gene 18s rRNA. (B) The fibroblasts were cultured in complete medium for the indicated time points post serum-starvation. Whole cell lysates were subjected to immunoblot analysis of PINK1 protein. β-actin was used as loading control. (C) Representative immunoblots depicting mitophagy related proteins, PINK1, Parkin, BNIP3, BNIP3L, NDP52, and optineurin in control and S-As fibroblasts. β-actin was used as loading control. (D) The fibroblasts were cultured in the presence of lysosomal protease inhibitors, E64d and pepstatin A, for 6 hours post serum-starvation. Cell lysates were subjected to immunoblot analysis of PINK1. β-actin was used as loading control. (E) The fibroblasts were cultured in DMEM complete medium for 48 hours post serum-starvation for immunofluorescence measurements at baseline. Representative images depicting control and S-As fibroblasts transduced with LC3B-GFP (green) and immunostained with PINK1-PE (red). (F) Representative images depicting fluorescent staining of mitochondria using MitoTracker Green (green) and lysosomes using LysoTracker Deep Red (red). Graphical data are represented as mean ± SEM from 3–4 independent experiments with at least 3 unique donors in each group. $^*p < 0.05$, $^{**}p < 0.01$, determined by unpaired two-tailed Student t-test.

PINK1 protein has a short half-life and its processing depends on the state of mitochondrial membrane potential (ΔΨm). Rapid and constitutive voltage-dependent degradation of PINK1 occurs in healthy cells whereas a loss in ΔΨm stabilizes the mitochondrial accumulation of full-length PINK1 [27]. In order to assess PINK1 processing and its stability in severe asthma, the levels of PINK1 protein was determined in S-As and control fibroblasts at 12 and 24 hours using western blot analysis. Interestingly, the full-length precursor form of PINK1 (~60kDa) was predominant in the S-As fibroblasts with faintly detectable bands of ~30-50kDa cleaved fragments (Fig 3B). In contrast, both full-length PINK1 and its cleaved isoforms were detected in the control fibroblasts at 12 hours, while the cleaved forms predominated at 24 hours (Fig 3B). Since mitochondrial integrity is critical for PINK1 processing, accumulation of endogenous full-length PINK1 at steady state in S-As fibroblasts reflects the dissipation of ΔΨm.

Since the mitophagy machinery involves the coordinated activities of multiple proteins, including PINK1, Parkin, BCL2/adenovirus E1B-interacting protein 3-like (BNIP3L)/ Nix, BNIP3, Optineurin and NDP52 [28], we also assessed their expression by western blot analysis. In addition to PINK1 and Parkin, the protein expression of several other mitophagy markers, including BNIP3, BNIP3L, Optineurin and NDP52 was also increased in S-As

fibroblasts in comparison to control fibroblasts (Fig 3C), suggesting the activation of the PINK1/Parkin-mediated mitophagy machinery that delivers the damaged mitochondria to autophagosomes.

To confirm the observed increase in mitophagy in S-As fibroblasts, we studied the lysosomal turnover of PINK1 by culturing the fibroblasts in the presence of E64d and pepstatin A for 6 hours. In contrast to control fibroblasts, significant accumulation of the full-length precursor form of PINK1 was observed in S-As fibroblasts upon treatment with E64d and pepstatin A (Fig 3D), indicating their lysosomal turnover through autophagy. The fibroblasts were further transduced with LC3B-GFP and stained for PINK1 to ascertain PINK1 co-localization with autophagosomes. The intensity of LC3B and PINK1 immunofluorescence was brighter in the S-As fibroblasts when compared to control (Fig 3E). Moreover, higher co-localization of PINK1 with LC3B was indicated proving increased mitophagy levels in these fibroblasts. This was further confirmed using MitoTracker Green which specifically stains mitochondria and LysoTracker Deep Red that stains lysosomes. A large proportion of the mitochondria in S-As fibroblasts was found to co-localize with lysosomes indicating their degradation via the lysosomal pathway (Fig 3F). Taken together, these findings reflect the presence of damaged mitochondria in S-As fibroblasts that triggers the stabilization of PINK1 and subsequent activation of PINK1/Parkin-mediated mitophagy machinery.

## 4. Enhanced mitochondrial depolarization in severe asthmatic fibroblasts

The observed increase in mitophagy flux in severe asthmatic fibroblasts is suggestive of mitochondrial depolarization. The state of mitochondrial polarization in bronchial fibroblasts was therefore, determined using the fluorescent JC-1 probe. Mitochondrial JC-1 aggregate fluorescence was reduced by 25% in S-As fibroblasts when compared to control fibroblasts ($p = 0.0248$) (Fig 4A) indicating that the $\Delta\Psi$m is partially depolarized in these fibroblasts.

The ability of mitochondrial dehydrogenases in metabolically active cells to convert MTT to a colored formazan product could be used as an index of mitochondrial activity [29, 30]. Across both 24 and 48 hours, the S-As fibroblasts displayed significantly reduced mitochondrial metabolic activity compared to the control fibroblasts (Fig 4B). Taken together, the concomitant reduction in $\Delta\Psi$m and mitochondrial metabolic activity suggest compromised mitochondrial function in the severe asthmatic fibroblasts.

## 5. Metabolic adaptation of severe asthmatic fibroblasts through activation of AMPKα

Mitochondrial dysfunction is known to induce adaptive changes to overcome adverse cellular consequences and cell death [31, 32]. Mitochondrial dysfunction in primary human fibroblasts triggered an adaptive survival mechanism requiring AMP-activated protein kinase (AMPK)-α activation and glucose supply [33]. We, therefore, assessed the phosphorylation status of AMPKα in S-As fibroblasts. The S-As fibroblasts displayed remarkable amplification in the p-AMPKα/AMKPα ratio ($p = 0.0029$) (Fig 5A) indicating an active coping mechanism in these fibroblasts in response to mitochondrial insult.

Moreover, AMPK triggers the activation of NAD+-dependent type III deacetylase sirtuin 1 (SIRT1) and its downstream peroxisome proliferator-activated receptor-γ coactivator 1α (PGC1α) leading to induction of mitochondrial gene expression [34, 35]. The increased AMPKα phosphorylation was accompanied by an increase in the expression of SIRT1 ($p = 0.0303$) and PGC1α ($p = 0.0035$) in the S-As fibroblasts when compared to control (Fig 5B and 5C), indicating an upregulation in mitochondrial biogenesis. Furthermore, the mitochondrial mass within the two group of fibroblasts was measured using the MitoTracker probe

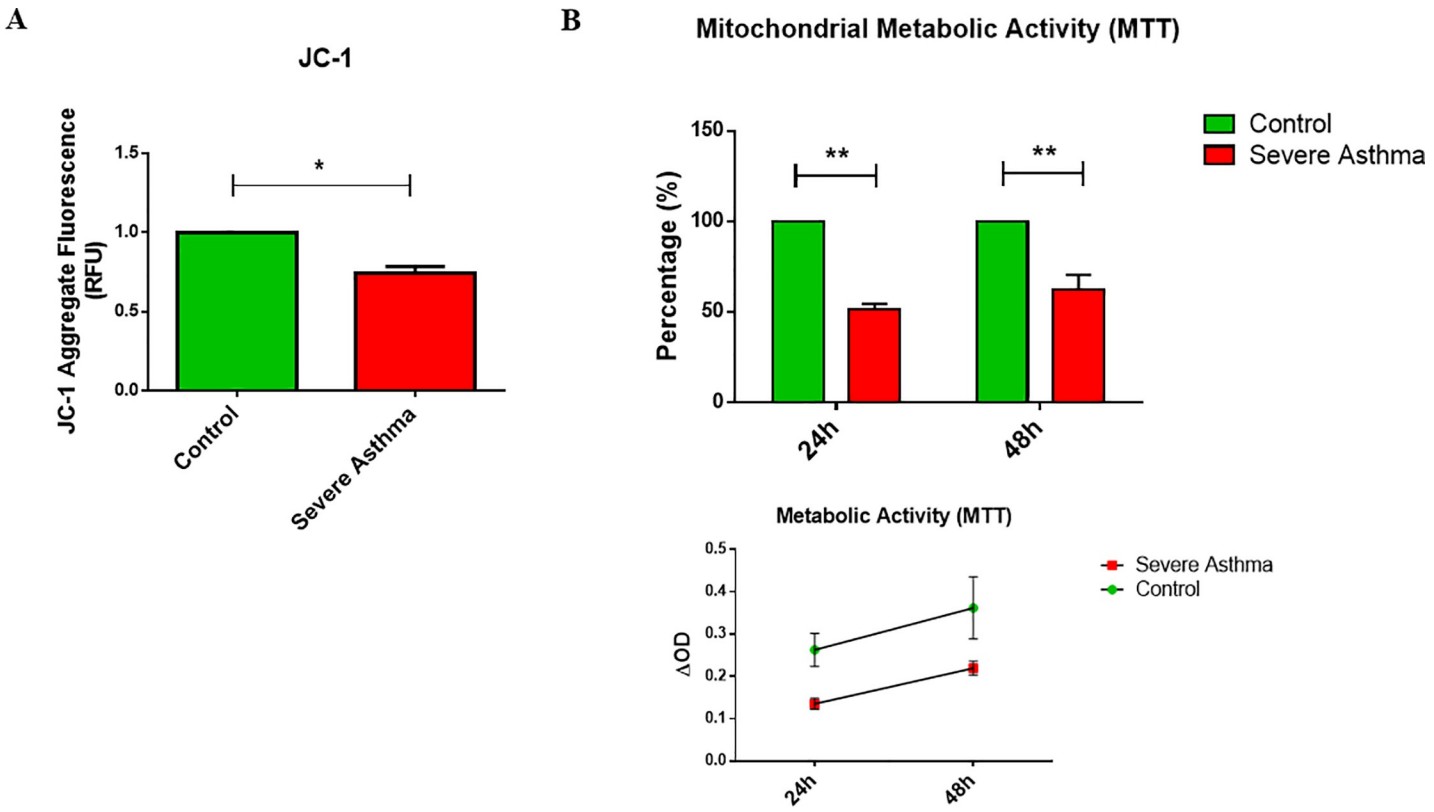

**Fig 4. Mitochondrial function is impeded in severe asthmatic fibroblasts. (A)** The control and S-As fibroblasts were labelled with JC-1 dye for 10 minutes and cultured thereafter for 4 hours. **(B)** The control and S-As fibroblasts were cultured in complete medium for up to 48 hours post serum-starvation. MTT reagent was added at the indicated time points and spectrophotometric readings were taken after 3 hours of incubation. Graphical data are represented as mean ± SEM relative to the control and representative of two independent experiments with each condition performed in triplicate. $^{*}p < 0.05$, $^{**}p < 0.01$, determined using unpaired two-tailed Student t-test.

and comparable mitochondrial content was observed between the severe asthmatic and control fibroblast population (Fig 5D), suggesting an efficient turnover of depolarized mitochondria in the severe asthmatic fibroblasts.

In order to ensure the turnover of defective mitochondria in S-As fibroblasts, we further measured cellular apoptosis and senescence in these fibroblasts by Annexin V and β-galactosidase staining, respectively. No signs of cellular apoptosis or senescence were observed in these cells (Fig 5E and 5F). These results demonstrated that mitochondrial depolarization in S-As fibroblasts induced an adaptive survival mechanism through AMPKα phosphorylation, and SIRT1 and PGC1α activation.

## 6. Increased pro-fibrotic and pro-inflammatory signaling in severe asthmatic fibroblasts

In the presence of these mitochondrial alterations, we next wanted to determine the functional phenotype of these bronchial fibroblasts. Fibroblasts from asthmatic airways are known to exhibit a pro-fibrotic and pro-inflammatory phenotype with excessive secretion of ECM proteins, cytokines and chemokines when compared to non-asthmatics [36–38]. Therefore, we characterized their baseline expression of ECM proteins, including collagen types I, III and V (*COL1A1*, *COL3A1 and COL5A1*) and fibronectin (*FN1*), cytokines, including *IL-6* and *IL-11*,

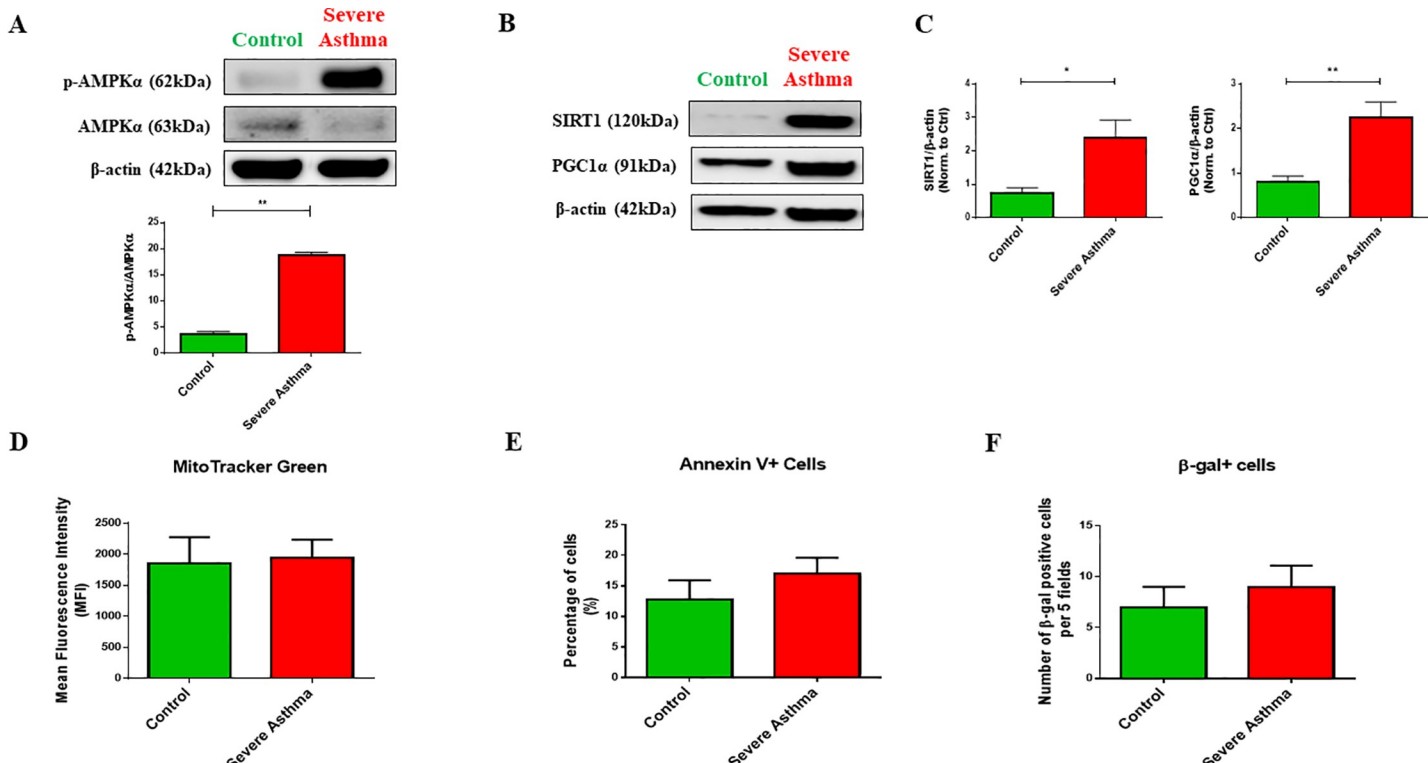

**Fig 5. Adaptive fibroblast persistence through AMPKα phosphorylation, SIRT1 and PGC1α expression in severe asthmatic fibroblasts. (A)** The fibroblasts were cultured in complete medium for 24 hours post serum-starvation. Whole cell lysates were subjected to immunoblot analysis of AMPKα and p-AMPKα (top panel). Densitometric analysis of AMPKα phosphorylation represented as the ratio of p-AMPKα to AMPKα (bottom panel). **(B)** Representative immunoblots depicting protein levels of SIRT1 and PGC1α in control and S-As fibroblasts. β-actin was used as loading control. **(C)** Densitometric analysis of SIRT1 and PGC1α in control and S-As fibroblasts. **(D)** The fibroblasts were cultured in complete medium for 48 hours post serum-starvation. Quantitative representation of MitoTracker Green fluorescence showing mitochondrial content in mean fluorescent intensity (MFI). **(E)** The levels of Annexin-V positive cells and **(F)** the levels of β-gal positive cells in control and S-As fibroblasts. Graphical data are represented as mean ± SEM from 2–4 independent experiments with at least 3 unique donors in each group. $^{*}p < 0.05$, $^{**}p < 0.01$, determined by unpaired two-tailed Student t-test.

and chemokines, including *IL-8* and *GROα (CXCL1)*, by qRT-PCR. The two groups of fibroblasts were distinct in their basal expression of ECM as well as cytokines. The S-As fibroblasts demonstrated increased basal expression of *COL1A1* (p = 0.0024), *COL3A1* (p = 0.0027) and *COL5A1* (p = 0.0249), compared to the healthy controls (Fig 6A). Although the S-As fibroblasts showed an increased trend in the mRNA expression of *IL-6* and *GROα*, it did not reach statistical significance (Fig 6B and 6C). However, the S-As fibroblasts demonstrated enhanced expression of *IL-11* (p = 0.0259) and *IL-8* (p = 0.0231) (Fig 6B and 6C). Consistent with previous reports [37, 38], the severe asthmatic fibroblasts demonstrated increased expression of pro-fibrotic and pro-inflammatory mediators and cytokines. Taken together, our results suggest that the heightened mitochondrial QC mechanisms of mitophagy and biogenesis promote a pro-fibrotic and pro-inflammatory phenotype in S-As fibroblasts.

Since autophagy has previously been reported to be a critical regulator of pro-fibrotic signaling in primary human atrial myofibroblasts [39], we targeted autophagy using the pharmacological agent, 3-MA, to block autophagosomal formation. Treatment with 3-MA effectively suppressed *COL1A1*, *COL3A1*, *COL5A1*, and *FN1* gene expression in both group of fibroblasts (Fig 6D). This indicated autophagy as a potentially targetable pathway to ameliorate fibrotic signaling in severe asthmatic fibroblasts.

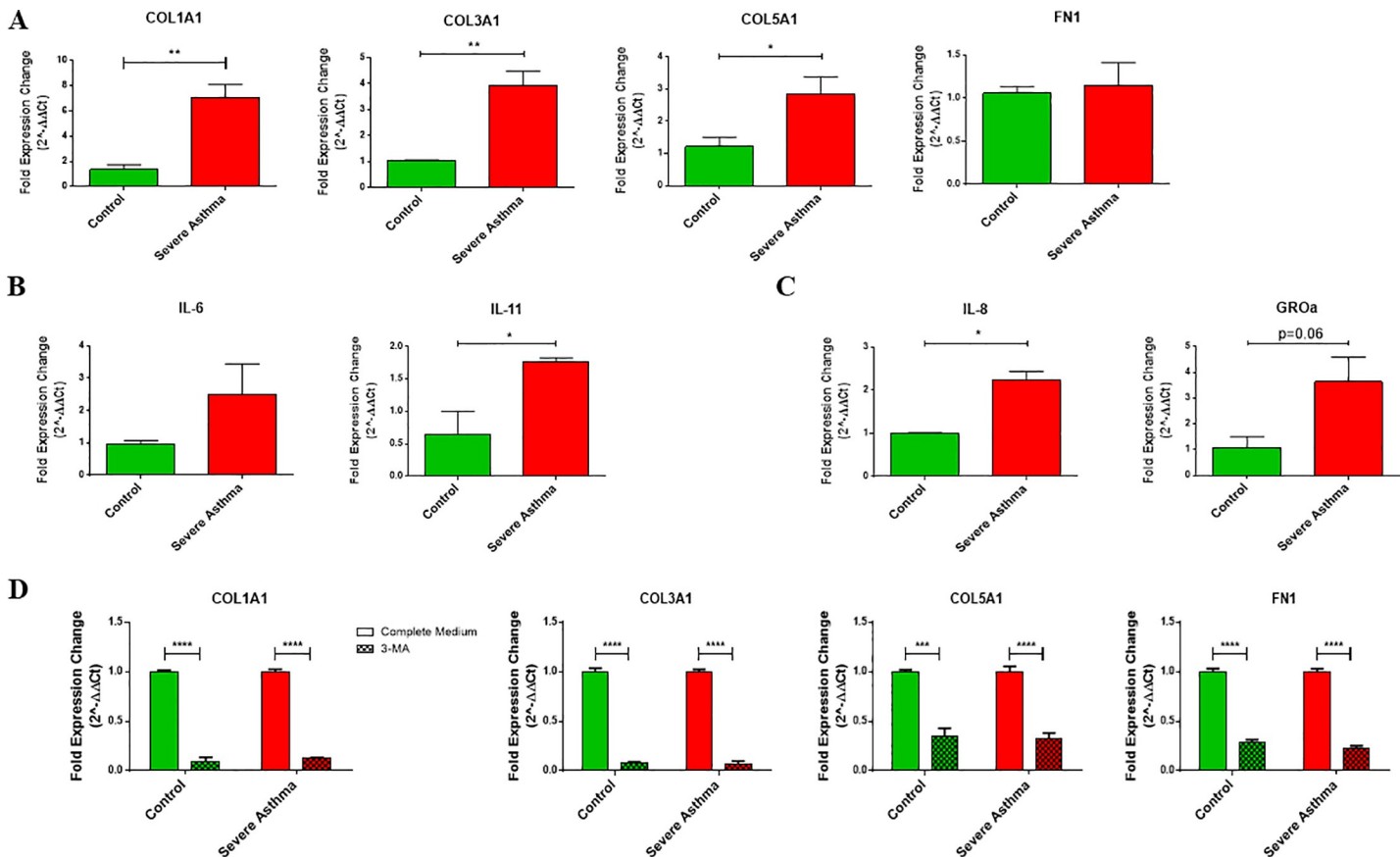

**Fig 6. Pro-fibrotic and pro-inflammatory signaling is increased in severe asthmatic fibroblasts.** The fibroblasts were cultured in DMEM complete medium for 2 hours post serum-starvation. Under basal conditions, mRNA expression of **(A)** ECM components *COL1A1*, *COL3A1*, *COL5A1* and *FN1*, **(B)** cytokines *IL-6* and *IL-11*, and **(C)** chemokines *IL-8* and *GROα*, in control and S-As fibroblasts was analysed by qRT-PCR and expressed as fold expression change relative to control fibroblasts post normalization to housekeeping gene 18s rRNA. **(D)** The fibroblasts were treated with autophagy inhibitor, 3-MA (1mM), for 48 hours, and the mRNA expression of ECM components *COL1A1*, *COL3A1*, *COL5A1* and *FN1*, in control and S-As fibroblasts was analysed by qRT-PCR and expressed as fold expression change relative to control fibroblasts post normalization to housekeeping gene 18s rRNA. Data are represented as mean ± SEM from at least 2 independent experiments. $^{*}p < 0.05$, $^{**}p < 0.01$, $^{***}p < 0.001$, $^{****}p < 0.0001$, determined by unpaired two-tailed Student t-test (Control vs Severe Asthma) or unpaired t-test with multiple comparisons using the Holm-Sidak method (Complete Medium vs 3-MA).

## Discussion

In light of the increasing evidence of mitochondrial dysfunction in asthma, we investigated the role of mitophagy in enhancing fibroblast persistence in severe asthma. Here, by using an *ex vivo* cell culture approach, we provide evidence that fibroblasts from severe asthmatic patients exhibit substantial mitochondrial defects at baseline that is accompanied by increased mitophagy flux in comparison to healthy control fibroblasts. Nevertheless, these fibroblasts demonstrated cellular adaptation to mitochondrial insult by upregulating AMPKα phosphorylation, SIRT1 and PGC1α expression which prevented cellular apoptosis or senescence. Furthermore, the severe asthmatic fibroblasts exhibited a pro-fibrotic and pro-inflammatory phenotype despite their intrinsic mitochondrial alterations. These results have important implications in understanding the role of mitochondrial QC in promoting sub-epithelial fibrosis in severe asthma.

Autophagy acts as a sentinel responsible for organelle quality control [40]. Constant turnover of damaged mitochondria is a prerequisite for the establishment of functional mitochondrial network. Since the mitochondria in severe asthma are constantly exposed to oxidative

damage [41, 42], enhanced mitophagy levels may serve as a stress adaptation mechanism to avoid cell death. Bioinformatics analysis indicated enrichment of genes associated with the mitophagy pathway in asthmatic fibroblasts (Fig 1C) and in accordance, the mitophagy genes, *PINK1* and *PRKN*, were upregulated in severe asthmatic fibroblasts (Fig 3A).

PINK1 processing depends on the mitochondrial membrane potential and its accumulation in the precursor form is indicative of enhanced mitophagy [27]. PINK1 was detectable in the full-length and cleaved forms in control fibroblasts, while the full-length precursor was predominant in severe asthmatic fibroblasts (Fig 3B). The expression of a variety of mitophagy related proteins, including Parkin, BNIP3, BNIP3L, Optineurin and NDP52, that are involved in varying stages of mitophagy [43, 44], was also upregulated in severe asthmatic fibroblasts (Fig 3C). Increased LC3B lipidation and p62 levels were also observed (Fig 2C and 2D), indicating enhanced mitophagy levels in severe asthmatic fibroblasts.

Generally, p62 levels are inversely correlated to the activation of autophagy [25]. The increased p62 protein expression noted in severe asthmatic fibroblasts in comparison to control fibroblasts (Fig 2C), however, aligns with some other studies where upregulation of p62 levels was observed with an increase in autophagy flux [45–47]. This could perhaps be explained by the fact that p62 is not specific only to the autophagy pathway and may be dispensable for the execution of parkin-mediated mitophagy [48]. Furthermore, p62 expression levels were found to depend on the rate of autophagic degradation, transcriptional upregulation, and availability of lysosomal-derived amino acids, indicating that they may not always inversely correlate with autophagic activity [49]. Hence, we looked at multiple autophagy and mitophagy markers in this study. The elevated gene and protein expression of LAMP2 (Fig 2B–2D), which is essential for the fusion between autophagic vacuoles and lysosomes [50], further implied constitutive activation of autophagy in these fibroblasts. Increased formation of autophagosomes and increased lysosomal activity (Fig 2A–2E) were also observed using immunofluorescent staining confirming the lysosomal delivery of autophagic vacuoles in severe asthmatic fibroblasts.

Static levels of LC3B provide an incomplete assessment of autophagy without the evaluation of autophagy flux. The accumulation of LC3BII and p62 in the presence of E64d and pepstatin A, which inhibit lysosomal proteolytic activity [25], was more prominent in severe asthmatic fibroblasts (Fig 2F). In control fibroblasts, while an accumulation of LC3BII was noted with E64d and pepstatin A treatment, the lack of p62 accumulation with treatment may be explained by the short incubation time of 6 hours, further stressing on the distinct phenotypes exhibited by the control and severe asthmatic fibroblasts. This differential kinetic regulation of LC3B and p62 by small molecules has been reported previously where a faster LC3B response to treatment was observed following which p62 levels increased at a later time point [51]. However, their increase in severe asthmatic fibroblasts indicates their transcriptional upregulation as well confirming an increase in autophagy flux in severe asthmatic fibroblasts when compared to control fibroblasts. Additionally, treatment with E64d and pepstatin A also led to increased accumulation of the full-length precursor form of PINK1 in the severe asthmatic fibroblasts (Fig 3D) confirming active turnover of these proteins through mitophagy. Furthermore, the increased co-localizations of mitochondria with autophagosomes as well as lysosomes substantiated the increase in mitophagy flux in S-As fibroblasts (Fig 3E and 3F).

Dissipation of ΔΨm is a sign of mitochondrial damage leading to the accumulation of PINK1 in severe asthmatic fibroblasts. In comparison with control fibroblasts, severe asthmatic fibroblasts were observed with a 25% reduction in ΔΨm (Fig 4A) and a reduction in mitochondrial metabolic activity (Fig 4B). Despite their intrinsic mitochondrial defects, the severe asthmatic fibroblasts demonstrated increased pro-fibrotic ECM (Fig 6A), cytokine (Fig 6B) and chemokine (Fig 6C) gene expression. The ability of these severe asthmatic fibroblasts

to increasingly produce ECM proteins as well as pro-fibrotic and pro-inflammatory cytokines support their active role in promoting remodeling and inflammation, two key processes involved in the pathogenesis of asthma. Our observations were in line with previous reports showing that oxidative stress-induced mitochondrial dysfunction drives inflammation and airway smooth muscle remodeling in COPD patients [52]. Additionally, autophagy was recently reported to be a critical regulator of asthmatic airway remodeling [53]. In agreement, treatment of bronchial fibroblasts with well-known autophagy inhibitor 3-MA significantly reduced *COL1A1*, *COL3A1*, *COL5A1* and *FN1* gene expression in both non-asthmatic and S-As fibroblasts (Fig 6D). Autophagy was previously shown to regulate TGF-β1-induced fibrotic response in primary human airway smooth muscle cells [54], indicating autophagy as a critical pathway for ECM secretion in airway smooth muscle cells. The positive correlation between the gene expression of *ATG5* and *COL5A1* in the airways of refractory asthmatic subjects further suggests that dysregulation of autophagy may contribute to fibrosis in asthmatic airways, particularly in difficult-to-treat refractory asthmatic individuals [18].

At the cellular level, mitochondrial dysfunction triggers adaptive changes to overcome unfavorable cellular consequences and escape cell death [31, 32]. These adaptive mechanisms include induction of mitophagy and mitochondrial biogenesis, alterations in mitochondrial dynamics and morphology, upregulation of glycolysis and alterations in anti-oxidant responses [55]. AMPK is a key nutrient and energy sensor that is central to the regulation of cellular energy homeostasis [56]. AMPK is usually activated as a rescue mechanism when the intracellular AMP to ATP ratio is high [56]. AMPK impacts metabolism and growth by regulating a number of pathways including glucose and lipid metabolism, autophagy and cell polarity. In a study by Distelmaier *et al.*, primary human fibroblasts exhibiting Complex I-associated mitochondrial dysfunction was found to heavily rely on extracellular glucose and AMPKα phosphorylation to initiate an adaptive cell survival response [33]. The mitochondrial depolarization exhibited by the severe asthmatic fibroblasts was accompanied by the increased phosphorylation of AMPKα (Fig 5A) which enabled the cells to metabolically adapt to the innate mitochondrial damage without any induction of cellular apoptosis or senescence (Fig 5E and 5F). Furthermore, elevation in NAD+ levels also activate SIRT1 that in coordination with AMPK regulates mitochondrial mass, ATP production and nutrient oxidation, with the help of transcription co-factor, PGC1α [35, 57]. Accordingly, increased SIRT1 and PGC1α expression was noted in the severe asthmatic fibroblasts (Fig 5B and 5C). Furthermore, there was no significant change in the mitochondrial content between the severe asthmatic and control fibroblasts (Fig 5D). Taking into consideration the increased mitophagy levels in severe asthmatic fibroblasts, the comparable mitochondrial content within the two groups suggest unimpeded mitochondrial turnover in the diseased fibroblasts.

AMPKα deficiency has previously been reported to compromise SIRT1 activity resulting in impaired PGC1α deacetylation, thereby impeding mitochondrial gene expression [35]. Similarly, AMPK inhibition was also found to induce cell cycle arrest and apoptosis [58]. Since mitochondrial dysfunction in primary human fibroblasts triggered an adaptive survival mechanism requiring AMPKα activation [33], the activation of the AMPKα/Sirt1/PGC1α signaling axis in severe asthmatic fibroblasts led us to conclude that AMPK appears to act as the trigger for mitochondrial damage-induced metabolic adaptation in severe asthmatic fibroblasts.

Generally, cells rely on the highly efficient mitochondrial oxidative phosphorylation (OXPHOS) as the primary source of cellular energy. However, human cancer cells switch the core cellular energy source from mitochondrial OXPHOS to aerobic glycolysis, a phenomenon termed Warburg effect. A metabolic shift to aerobic glycolysis was shown to predispose fibroblasts to a pro-fibrotic phenotype [59, 60]. TGF-β1-induced mitochondrial biogenesis and aerobic glycolysis regulated myofibroblast contractility and differentiation [59]. Likewise,

glycolytic reprogramming contributed to the pathogenesis of lung fibrosis by promoting myo-fibroblast differentiation [60]. It would, therefore, be interesting to compare the glycolytic potential and oxidative phosphorylation capacity in these severe asthmatic fibroblasts in future studies.

Although the precise mechanistic signaling of the observed mitophagy levels remain elusive, it is plausible that in the presence of depolarized mitochondria in severe asthmatic fibroblasts, mitophagy is elevated as a rescue mechanism against mitochondrial oxidative stress to prevent apoptosis and senescence in bronchial fibroblasts. This, in turn, may exacerbate the pathophysiology of severe asthma through the persistence of these fibroblasts. Further studies are necessary to elucidate the molecular events driven by these mitochondrial defects in severe asthmatic fibroblasts that lead to their persistent activation and fibrotic behavior.

## Conclusions

In summary, we show that increased mitophagy flux is associated with increased mitochondrial biogenesis and improved survival in S-As fibroblasts. As shown in Fig 7, we propose the following model of mitochondrial homeostasis in severe asthmatic fibroblasts. In the increased presence of oxidative stressors such as allergens, pollutants or cigarette smoke, the mitochondria are increasingly prone to damage in severe asthmatic fibroblasts. These damaged

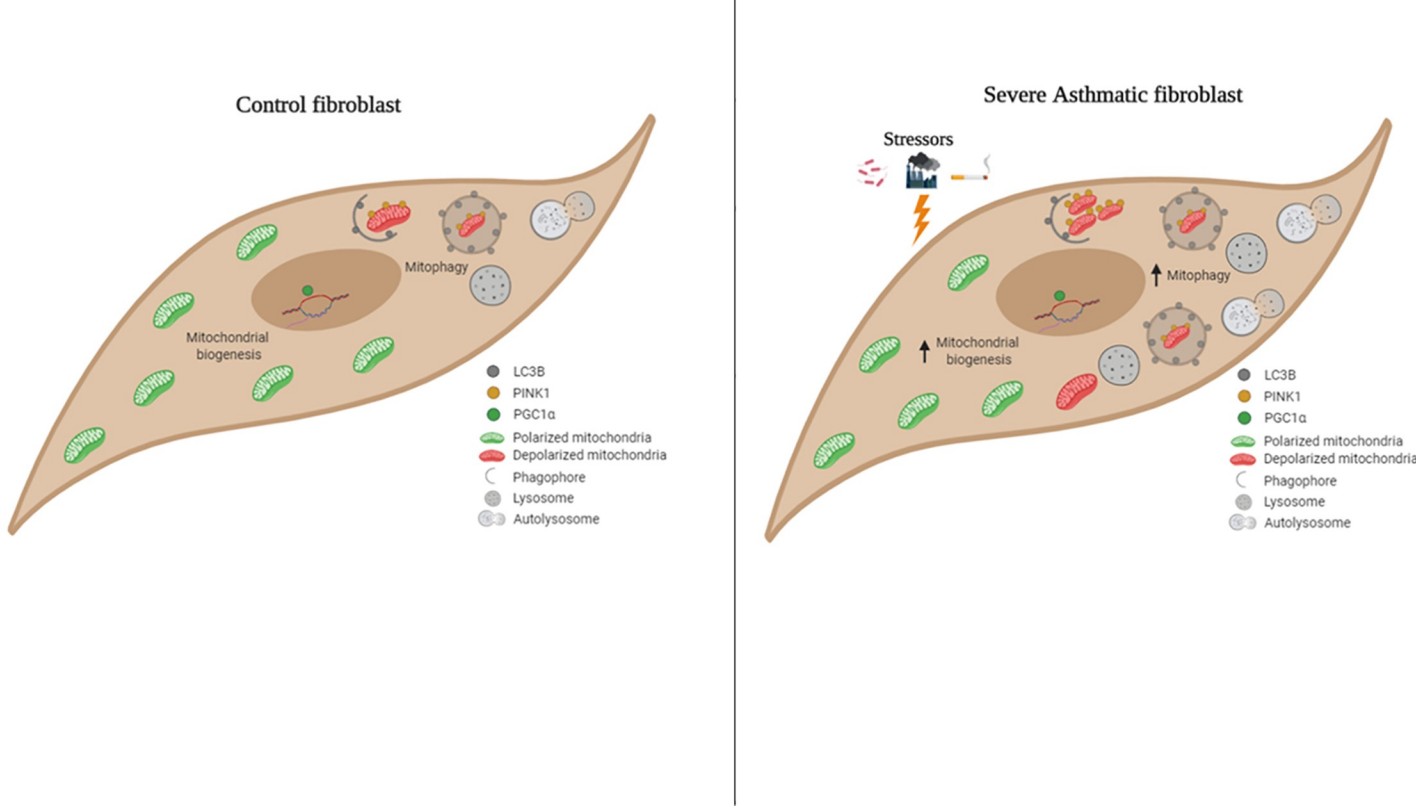

**Fig 7. Graphical abstract–schematic representation of mitochondrial homeostasis in control and severe asthmatic bronchial fibroblasts.** In control fibroblasts, mitochondrial homeostasis is ensured by basal levels of mitophagy and mitochondrial biogenesis. With exposure of hyperresponsive airways in severe asthma to stressors such as allergens, pollutants or cigarette smoke, the mitochondria are increasingly prone to damage in S-As fibroblasts. The damaged mitochondria are effectively recycled by increased mitophagy and biogenesis. The new mitochondria eventually become vulnerable to the mitochondrial stressors reflecting a vicious pathological cycle ensuring the increased persistence of S-As fibroblasts.

mitochondria are effectively cleared by increased mitophagy. To compensate for the mito-chondrial loss, new functional mitochondria are generated which eventually become vulnera-ble to the mitochondrial stressors. This reflects a vicious pathological cycle ensuring the increased persistence of severe asthmatic fibroblasts. Restoring mitochondrial integrity may help maintain a normal fibroblast phenotype and thus, attenuate the development of sub-epi-thelial fibrosis in severe asthmatic patients.

## Supporting information

**S1 Raw images.**
(PDF)

## Acknowledgments

We would like to acknowledge Dr. Saba Al Heialy and Dr. Thenmozhi Venkatachalam for their expertise, technical help and valuable support. We would also like to thank Dr. Abdul Wahid Ansari and Manju Jayakumar for their expertise in flow cytometry.

## Author Contributions

**Conceptualization:** Rakhee K. Ramakrishnan, Qutayba Hamid.

**Data curation:** Rakhee K. Ramakrishnan, Andrea K. Mogas.

**Formal analysis:** Rakhee K. Ramakrishnan, Khuloud Bajbouj, Mahmood Y. Hachim, Rabih Halwani.

**Funding acquisition:** Bassam Mahboub, Rifat Hamoudi, Rabih Halwani, Qutayba Hamid.

**Investigation:** Rakhee K. Ramakrishnan, Rabih Halwani.

**Methodology:** Rakhee K. Ramakrishnan, Khuloud Bajbouj, Mahmood Y. Hachim, Rifat Hamoudi, Rabih Halwani.

**Project administration:** Qutayba Hamid.

**Resources:** Andrea K. Mogas, Bassam Mahboub, Ronald Olivenstein, Rifat Hamoudi, Qutayba Hamid.

**Software:** Mahmood Y. Hachim, Rifat Hamoudi.

**Supervision:** Bassam Mahboub, Rabih Halwani, Qutayba Hamid.

**Validation:** Rakhee K. Ramakrishnan.

**Visualization:** Rakhee K. Ramakrishnan.

**Writing – original draft:** Rakhee K. Ramakrishnan.

**Writing – review & editing:** Khuloud Bajbouj, Mahmood Y. Hachim, Bassam Mahboub, Rifat Hamoudi, Rabih Halwani, Qutayba Hamid.

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
