## [Decision Letter · Decision Letter 0]

26 Jun 2020

PONE-D-20-12146

Enhanced mitophagy in bronchial fibroblasts from severe asthmatic patients

PLOS ONE

Dear Dr. Hamid,

Thank you for submitting your manuscript to PLOS ONE. After careful consideration, we feel that it has merit but does not fully meet PLOS ONE’s publication criteria as it currently stands. Therefore, we invite you to submit a revised version of the manuscript that addresses the points raised during the review process.

In addition to considering the reviewers’ comments displayed below, the authors need to resolve the following issues:

1. It is standard for RT-qPCR that at least two different reference genes are used. What precautions have been taken to ensure that 18sRNA is indeed a valid reference gene in this particular experimental setting?

2. It is mentioned in figure legends that the fibroblasts were serum starved before incubated in the complete cell culture medium. The rationale for this approach should be clearly stated and the complete protocol for fibroblast cultivation should be described in Materials and methods.  

3. The autophagic flux experiments in Fig. 1 are not convincing. Namely, the increase in SQSTM1/p62 protein levels in the absence of transcriptional upregulation actually supports the inhibition, rather than increase in autophagic flux. The results obtained with E64/pepstatin further add to the confusion, as they seem to reduce SQSTM1/p62 levels, which does not make sense if its proteolysis was efficiently blocked. Finally, the small increase in LC3-II levels in E64/pepstatin-incubated asthmatic fibroblast compared to untreated ones seems to correlate with the increased protein loading (as judging from actin levels), which is again incompatible with the increase in autophagic flux. The statistical analysis of the Fig. 2F should help in resolving these issues.

4. The phospho-AMPK/AMPK immunoblots should be shown.   

Reviewer 1:

Rakhee et al describe a role for mitophagy in bronchial fibroblasts from severe asthmatic patients. Enhanced mitophage may contribute to fibrosis in the fibrosis of severe asthma. The data presented are convincing and the story is integrated. There are however several issues that need to be addressed.

1. For bioinformatics analysis in Fig 1, the authors should show the heatmap of differential gene expression.

2. In the analysis of co-localization in Fig 2 and 3, showing the merged image will be more convincing. In Fig 3B,the total PINK1 in control seems to be more than that in S-As. The authors should explain this result. Since the Pink/Parkin signal enhanced in S-As, is there a difference in the ubiquitination of mitochondrial proteins? If the ubiquitination of mitochondrial proteins enhances in S-As, Parkin, as an E3 ubiquitinligase, what is the target protein. 

3. In Fig 5,activation of AMPKa induced the upregulation of SIRT1 and PGC1a in S-As. The authors should inhibit AMPKα to assess the change of SIRT1, PGC1α and the apoptosis of fibroblast in S-As, thus providing more solid evidence to support their conclusion.

4. In Fig 6, the authors suggest that the enhanced turnover of damaged mitochondria in S-As may contribute to fibrosis in severe asthma by promoting the persistence and pro-fibrotic phenotype of fibroblasts. The evidence is not sufficient. The author should show whether inhibitors of mitophagy/autophagy or the blocking of Pink/Parkin signaling affects these pro-fibrotic phenotype of fibroblasts.

Reviewer 2:

I have reviewed your manuscript “Enhanced mitophagy in bronchial fibroblasts from severe asthmatic patients”. It is of interest, and I am able to consider it for publication in its current form. The authors report that in severe asthmatic fibroblasts, the differential expression of mitophagy genes, PINK1 and PRKN, were accompanied by the accumulation of PINK1, Parkin and other mitophagy proteins. Also, accumulation of endogenous LC3B-Ⅱ, p62 and PINK1 in severe asthmatic fibroblasts was observed. These fibroblasts displayed neither an apoptotic nor senescent phenotype but a pro-fibrotic phenotype. Interestingly, whether these pro-fibrotic fibroblasts are able reverse to pre-epithelial mesenchymal transformation state by improving mitophagy flux. Study needs to clearly demonstrate the relationship between turnover of damaged mitochondria and epithelial mesenchymal transformation, which supports the hypothesis that a role for mitophagy can be attributed to bronchial remodeling in severe asthmatic patients.

We look forward to receiving your revised manuscript.

Kind regards,

Vladimir Trajkovic

Academic Editor

PLOS ONE

Journal Requirements:

2. Thank you for including your ethics statement:  "The primary bronchial fibroblasts used in our study were obtained from the Quebec Respiratory Health Research Network Tissue Bank (McGill University Health Centre/ Meakins-Christie Laboratories Tissue Bank, Montreal, Canada). The original study was approved by institutional review board (2017-2581) and the subjects had provided written informed consent."   

3. Please provide the catalog numbers of the cell lines, dilutions of all antibodies, and information on secondary antibodies used in your study.

4. We noted in your submission details that a portion of your manuscript may have been presented or published elsewhere.

"This work was accepted as a poster presentation at the CHEST Congress 2019."

7. PLOS requires an ORCID iD for the corresponding author in Editorial Manager on papers submitted after December 6th, 2016. Please ensure that you have an ORCID iD and that it is validated in Editorial Manager. To do this, go to ‘Update my Information’ (in the upper left-hand corner of the main menu), and click on the Fetch/Validate link next to the ORCID field. This will take you to the ORCID site and allow you to create a new iD or authenticate a pre-existing iD in Editorial Manager. Please see the following video for instructions on linking an ORCID iD to your Editorial Manager account: https://www.youtube.com/watch?v=_xcclfuvtxQ

8. Your ethics statement must appear in the Methods section of your manuscript. If your ethics statement is written in any section besides the Methods, please move it to the Methods section and delete it from any other section. Please also ensure that your ethics statement is included in your manuscript, as the ethics section of your online submission will not be published alongside your manuscript.

Reviewers' comments:

Reviewer's Responses to Questions

**Comments to the Author**

1. Is the manuscript technically sound, and do the data support the conclusions?

Reviewer #1: Yes

Reviewer #2: Yes

2. Has the statistical analysis been performed appropriately and rigorously? 

Reviewer #1: Yes

Reviewer #2: Yes

3. Have the authors made all data underlying the findings in their manuscript fully available?

Reviewer #1: Yes

Reviewer #2: Yes

4. Is the manuscript presented in an intelligible fashion and written in standard English?

Reviewer #1: Yes

Reviewer #2: Yes

5. Review Comments to the Author

Reviewer #1: Rakhee et al describe a role for mitophagy in bronchial fibroblasts from severe asthmatic patients. Enhanced mitophage may contribute to fibrosis in the fibrosis of severe asthma. The data presented are convincing and the story is integrated. There are however several issues that need to be addressed.

1. For bioinformatics analysis in Fig 1, the authors should show the heatmap of differential gene expression.

2. In the analysis of co-localization in Fig 2 and 3, showing the merged image will be more convincing. In Fig 3B，the total PINK1 in control seems to be more than that in S-As. The authors should explain this result. Since the Pink/Parkin signal enhanced in S-As, is there a difference in the ubiquitination of mitochondrial proteins? If the ubiquitination of mitochondrial proteins enhances in S-As, Parkin, as an E3 ubiquitinligase, what is the target protein.

3. In Fig 5，activation of AMPKa induced the upregulation of SIRT1 and PGC1a in S-As. The authors should inhibit AMPKα to assess the change of SIRT1, PGC1α and the apoptosis of fibroblast in S-As, thus providing more solid evidence to support their conclusion.

4. In Fig 6, the authors suggest that the enhanced turnover of damaged mitochondria in S-As may contribute to fibrosis in severe asthma by promoting the persistence and pro-fibrotic phenotype of fibroblasts. The evidence is not sufficient. The author should show whether inhibitors of mitophage or the blocking of Pink/Parkin signaling affects these pro-fibrotic phenotype of fibroblasts.

Reviewer #2: There was no other concerns about research ethics, publication ethics and others.

6. PLOS authors have the option to publish the peer review history of their article (what does this mean?). If published, this will include your full peer review and any attached files.

Reviewer #1: No

Reviewer #2: **Yes: **Ching-Yuang Lin

---

## [Author Response · Author response to Decision Letter 0]

9 Sep 2020

Editor:

We would like to thank the Academic Editor and Editorial Office for their valuable comments and suggestions to help improve our manuscript. We hope we have addressed these concerns appropriately. Please find below our response to the queries.

1. It is standard for RT-qPCR that at least two different reference genes are used. What precautions have been taken to ensure that 18sRNA is indeed a valid reference gene in this particular experimental setting?

We agree that some studies use two or more housekeeping genes, however, in this case we wanted to select housekeeping genes that show minimum variation and also, have high expression across the samples. We tried different genes, including GAPDH, but they showed higher variation or naturally had low expression. Therefore, we stayed with 18s rRNA alone, based on the following criteria:

1) 18S rRNA gave consistently low Ct values (in the range of 8.7-9 across the fibroblasts included in this study) with stable high expression.

2) 18S rRNA had the least expression variability across samples. 

3) 18S is not involved in mitochondrial function and is therefore, a non-biased and reliable housekeeping gene in this study.

Literature search also indicated a study comparing the various commonly used housekeeping genes, including ACTB, GAPDH, 18S rRNA, and mitochondrial genes, that identified 18S rRNA as the most stable reference gene to normalize qRT-PCR data in primary human bronchial epithelial cells (Kuchipudi SV et al, 2012).

2. It is mentioned in figure legends that the fibroblasts were serum starved before incubated in the complete cell culture medium. The rationale for this approach should be clearly stated and the complete protocol for fibroblast cultivation should be described in Materials and methods.

Thank you for this suggestion. Since we used primary fibroblasts in our study, we noticed differences in their growth pattern irrespective of their diseased state. For instance, the primary bronchial fibroblasts exhibited doubling times varying from 30-48 hours. Therefore, in order to provide a more reproducible experimental condition and synchronize the population of proliferating cells across all the fibroblast cell types included in this study, serum-starvation was done prior to all experiments. We have made the appropriate changes in line 125.

3. The autophagic flux experiments in Fig. 1 are not convincing. Namely, the increase in SQSTM1/p62 protein levels in the absence of transcriptional upregulation actually supports the inhibition, rather than increase in autophagic flux. The results obtained with E64/pepstatin further add to the confusion, as they seem to reduce SQSTM1/p62 levels, which does not make sense if its proteolysis was efficiently blocked. Finally, the small increase in LC3-II levels in E64/pepstatin-incubated asthmatic fibroblast compared to untreated ones seems to correlate with the increased protein loading (as judging from actin levels), which is again incompatible with the increase in autophagic flux. The statistical analysis of the Fig. 2F should help in resolving these issues.

In addition to the increased LC3BII levels and LC3B lipidation in S-As fibroblasts (Fig. 2C), the concomitant increase in p62 levels prompted us to investigate the lysosomal function to exclude the possibility of lysosomal defect. This has been discussed in line 304. The observation of increased LAMP2 mRNA and protein expression coupled with enhanced LysoTracker fluorescence suggested increased lysosomal activity and thus, ruling out the possibility of defective autophagy clearance. The increased p62 protein expression noted in S-As fibroblasts in comparison to control fibroblasts (Fig. 2C), however, aligns with some other studies where upregulation of p62 levels was observed with an increase in autophagy flux (Colosetti P et al, 2009;Toepfer N et al, 2011; Zheng Q et al, 2011). This could also perhaps be explained by the fact that p62 is not specific only to the autophagy pathway and hence, we looked at multiple autophagy and mitophagy markers in this study. We have added this to line 530.

Further, we have also added the densitometric analysis of the blots in Fig. 2F to ascertain that the observed increase in LC3B, p62 and LAMP2 levels indicates increased autophagy flux in S-As fibroblasts.

4. The phospho-AMPK/AMPK immunoblots should be shown.

The immunoblots have been included in Fig. 5A.

Thank you again for your comments and we hope that we have satisfactorily addressed your concerns in the revised manuscript.

Reviewer 1:

We would like to thank you for your comments and suggestions. Due to the current restrictions posed by the COVID-19, we have tried our best in performing the required experiments. We believe we have addressed your concerns appropriately.

There are however several issues that need to be addressed.

1. For bioinformatics analysis in Fig 1, the authors should show the heatmap of differential gene expression.

Thank you for this suggestion. The heatmap has been added as Fig. 1A.

2. In the analysis of co-localization in Fig 2 and 3, showing the merged image will be more convincing. In Fig 3B, the total PINK1 in control seems to be more than that in S-As. The authors should explain this result. Since the Pink/Parkin signal enhanced in S-As, is there a difference in the ubiquitination of mitochondrial proteins? If the ubiquitination of mitochondrial proteins enhances in S-As, Parkin, as an E3 ubiquitin ligase, what is the target protein.

As requested, please find the merged images added to Figs. 2E, 3E and 3F.

Diseases such as asthma, COPD and IPF are known to possess phenotypically different fibroblasts that are responsible for the loss of the typical airway architecture and known to impair airway function. Therefore, we would expect the kinetics and stability of PINK1 processing to vary between the control and S-As fibroblasts. We believe this expression pattern is due to this difference. However, the accumulation of full-length PINK1 is a sign of mitochondrial damage and this distinct pattern was noticed only in S-As fibroblasts. Further, the PINK1 blots were repeated multiple times to ensure the greater accumulation of full-length PINK1 in S-As fibroblasts in comparison to controls.

PINK1 is known to phosphorylate Parkin activating its E3 ubiquitin ligase activity. The activated Parkin then poly-ubiquitylates mitochondrial substrates on the outer mitochondrial membrane (OMM), including mitofusin-1, -2, and VDAC1, thereby tagging the mitochondria for autophagic degradation (Gegg ME et al, 2010; Glauser S et al, 2011; Geisler S et al, 2010). Although we did not look at the ubiquitination status of these proteins per se, we investigated the expression of multiple adaptor proteins that serve as mitophagy receptors downstream of the ubiquitination to facilitate PINK1/Parkin-mediated mitophagy, including BNIP3L/Nix, BNIP3, p62, Optineurin and NDP52, which was found to be elevated in S-As fibroblasts in comparison to controls (Fig. 3C).

3. In Fig 5, activation of AMPKa induced the upregulation of SIRT1 and PGC1a in S-As. The authors should inhibit AMPKα to assess the change of SIRT1, PGC1α and the apoptosis of fibroblast in S-As, thus providing more solid evidence to support their conclusion.

The AMPK/Sirt1/PGC1α signaling axis is a well-known master regulator of metabolic/energetic homeostasis and is fine-tuned in response to different metabolic situations. We completely agree that the effects of AMPKα inhibition on SIRT1 and PGC1α signaling would provide us with more definitive answers, however, due to the current COVID-19 associated delays we were unable to perform this additional experiment. Nevertheless, we have discussed this more in line 592.

4. In Fig 6, the authors suggest that the enhanced turnover of damaged mitochondria in S-As may contribute to fibrosis in severe asthma by promoting the persistence and pro-fibrotic phenotype of fibroblasts. The evidence is not sufficient. The author should show whether inhibitors of mitophagy/autophagy or the blocking of Pink/Parkin signaling affects these profibrotic phenotype of fibroblasts.

Thank you for this recommendation. As suggested, we pharmacologically inhibited autophagy in these fibroblasts using the well-known inhibitor 3-MA and this inhibition was found to significantly downregulate the expression of ECM markers. This has been added to the Results section in line 497 and Fig. 6D, and appropriately discussed thereafter in line 560.

Thank you again for your comments and we hope that we have satisfactorily addressed your concerns in the revised manuscript.

Reviewer 2:

I have reviewed your manuscript “Enhanced mitophagy in bronchial fibroblasts from severe asthmatic patients”. It is of interest, and I am able to consider it for publication in its current form. The authors report that in severe asthmatic fibroblasts, the differential expression of mitophagy genes, PINK1 and PRKN, were accompanied by the accumulation of PINK1, Parkin and other mitophagy proteins. Also, accumulation of endogenous LC3B-Ⅱ, p62 and PINK1 in severe asthmatic fibroblasts was observed. These fibroblasts displayed neither an apoptotic nor senescent phenotype but a profibrotic phenotype. Interestingly, whether these pro-fibrotic fibroblasts are able reverse to pre-epithelial mesenchymal transformation state by improving mitophagy flux. Study needs to clearly demonstrate the relationship between turnover of damaged mitochondria and epithelial mesenchymal transformation, which supports the hypothesis that a role for mitophagy can be attributed to bronchial remodeling in severe asthmatic patients.

We would like to thank you for your comments and suggestions. Due to the current restrictions posed by the COVID-19, we have tried our best in performing some additional experiments. We hope we have addressed your concerns appropriately.

We agree that you have put forward an interesting line of thought. While mitochondrial dysfunction has been implicated as an important driver of EMT in cancer (Guerra F et al, 2017), autophagy is emerging as an important player in EMT-associated airway remodeling in asthma (Liu T et al, 2017). Previous work by our group has reported neutrophil driven inflammation in severe asthmatic patients to induce EMT in healthy bronchial epithelial cells (Haddad A et al, 2019), suggesting increased EMT in severe asthmatics as well. Therefore, it would definitely be interesting to investigate in future studies the involvement of mitophagy in driving this process in severe asthma. Nevertheless, we performed additional experiments using autophagy inhibitor, 3-MA, which was found to effectively diminish ECM expression across both control and S-As fibroblasts (Fig. 6D), highlighting the pathological role of autophagy in promoting subepithelial fibrosis in S-As patients. This has been added to the Results section in line 497 and Fig. 6D, and appropriately discussed thereafter in line 560. 

Thank you again for your comments and we hope that we have satisfactorily addressed your concerns in the revised manuscript.

Journal Requirements:

and

The manuscript has been adequately formatted.

2. Thank you for including your ethics statement: "The primary bronchial fibroblasts used in our study were obtained from the Quebec Respiratory Health Research Network Tissue Bank (McGill University Health Centre/ Meakins-Christie Laboratories Tissue Bank, Montreal, Canada). The original study was approved by institutional review board (2017-2581) and the subjects had provided written informed consent."

For additional information about PLOS ONE ethical requirements for human subjects research, please refer to

http://journals.plos.org/plosone/s/submission-guidelines#loc-human-subjects-research.

The ethics statement has been amended and included in the Methods section in line 118.

3. Please provide the catalog numbers of the cell lines, dilutions of all antibodies, and information on secondary antibodies used in your study.

The required information has been included.

4. We noted in your submission details that a portion of your manuscript may have been presented or published elsewhere.

"This work was accepted as a poster presentation at the CHEST Congress 2019."

The preliminary part of this study was accepted as an abstract for poster presentation at the CHEST Congress 2019. The work was, however, not peer-reviewed as the submission included an abstract alone. The abstract was published in the CHEST website and can be accessed using the below link:

https://journal.chestnet.org/article/S0012-3692(19)30215-6/fulltext

The current submitted manuscript includes increased sample size and additional experiments was performed to develop a more detailed understanding of the role of mitophagy in S-As bronchial fibroblasts.

5. We note that you have indicated that data from this study are available upon request. PLOS only allows data to be available upon request if there are legal or ethical restrictions on sharing data publicly. For more information on unacceptable data access restrictions, please see http://journals.plos.org/plosone/s/data-availability#loc-unacceptabledata-access-restrictions.

b) If there are no restrictions, please upload the minimal anonymized data set necessary to replicate your study findings as either Supporting Information files or to a stable, public repository and provide us with the relevant URLs, DOIs, or accession numbers. For a list of acceptable repositories, please see http://journals.plos.org/plosone/s/dataavailability#loc-recommended-repositories.

All data underlying the reported findings have been provided as part of the submitted article.

6. PLOS ONE now requires that authors provide the original uncropped and unadjusted images underlying all blot or gel results reported in a submission’s figures or Supporting Information files. This policy and the journal’s other requirements for blot/gel reporting and figure preparation are described in detail at https://journals.plos.org/plosone/s/figures#loc-blotand-gel-reporting-requirements and https://journals.plos.org/plosone/s/figures#loc-preparing-figures-from-image-files.

When you submit your revised manuscript, please ensure that your figures adhere fully to these guidelines and provide the original underlying images for all blot or gel data reported in your submission. See the following link for instructions on providing the original image data: https://journals.plos.org/plosone/s/figures#loc-original-images-for-blots-and-gels.

The required original blots have been included as a supporting informaiton file.

7. PLOS requires an ORCID iD for the corresponding author in Editorial Manager on papers submitted after December 6th, 2016. Please ensure that you have an ORCID iD and that it is validated in Editorial Manager. To do this, go to ‘Update my Information’ (in the upper left-hand corner of the main menu), and click on the Fetch/Validate link next to the ORCID field.

This will take you to the ORCID site and allow you to create a new iD or authenticate a pre-existing iD in Editorial Manager. Please see the following video for instructions on linking an ORCID iD to your Editorial Manager account:

https://www.youtube.com/watch?v=_xcclfuvtxQ

The ORCHID iD shall be updated.

8. Your ethics statement must appear in the Methods section of your manuscript. If your ethics statement is written in any section besides the Methods, please move it to the Methods section and delete it from any other section. Please also ensure that your ethics statement is included in your manuscript, as the ethics section of your online submission will not be published alongside your manuscript.

The ethics statement has been amended and included in the Methods section.

1) Please upload a new copy of Figures 1, 2, 3 and 6 as the detail is not clear. Please follow the link for more information:

https://blogs.plos.org/plos/2019/06/looking-good-tips-for-creating-your-plos-figures-graphics/

Thank you so much for your comments and for bringing this to our attention. We hope we have addressed these concerns appropriately. We have improved the scaling and sizing of our images to improve its details.

2) It appears that your ORCiD iD has not been validated in your Editorial Manager account and we are unable to proceed until that step is complete.

To validate your ORCiD iD in Editorial Manager, please follow the steps below:

i. In your Editorial Manager account, please go to ‘Update my Information’ (in the upper left-hand corner of the main menu), and click on the Fetch/Validate link next to the ORCiD field.

ii. This link will take you to the ORCiD site and allow you to create a new iD or authenticate a pre-existing iD in Editorial Manager.

For additional instructions, please watch the following video for a step-by-step demonstration:

https://www.youtube.com/watch?v=_xcclfuvtxQ

The ORCID iD has been updated in the Editorial Manager account. It is “0000-0002-1107-190X”.

---

## [Decision Letter · Decision Letter 1]

14 Oct 2020

PONE-D-20-12146R1

Enhanced mitophagy in bronchial fibroblasts from severe asthmatic patients

PLOS ONE

Dear Dr. Hamid,

Thank you for submitting your manuscript to PLOS ONE. After careful consideration, we feel that it has merit but does not fully meet PLOS ONE’s publication criteria as it currently stands. Therefore, we invite you to submit a revised version of the manuscript that addresses the points raised during the review process.

ACADEMIC EDITOR: While the study has been improved, the interpretation of p62 data needs additional attention. Namely, the unexpected finding that p62 levels were not increased in the presence of pepstatin/E64 could be explained by the relatively short incubation time (6 h). Also, the further increase in p62 levels in the presence of lysosomal proteolysis inhibitors indicates its transcriptional upregulation, which is consistent with its role in delivering autophagic cargo. The authors should briefly address these issues.   

We look forward to receiving your revised manuscript.

Kind regards,

Vladimir Trajkovic

Academic Editor

PLOS ONE

Reviewers' comments:

Reviewer's Responses to Questions

**Comments to the Author**

1. If the authors have adequately addressed your comments raised in a previous round of review and you feel that this manuscript is now acceptable for publication, you may indicate that here to bypass the “Comments to the Author” section, enter your conflict of interest statement in the “Confidential to Editor” section, and submit your "Accept" recommendation.

Reviewer #1: All comments have been addressed

2. Is the manuscript technically sound, and do the data support the conclusions?

Reviewer #1: Yes

3. Has the statistical analysis been performed appropriately and rigorously? 

Reviewer #1: Yes

4. Have the authors made all data underlying the findings in their manuscript fully available?

Reviewer #1: Yes

5. Is the manuscript presented in an intelligible fashion and written in standard English?

Reviewer #1: Yes

6. Review Comments to the Author

Reviewer #1: The manuscript by Rakhee et al. provides the key role of mitochondrial homeostasis in severe asthmatic fibroblasts. Enhanced mitophagy in bronchial fibroblasts from severe asthmatic patients stimulates the development of subepithelial fibrosis.

The manuscript is novel, and the experimental approaches are adequate, and the results shown support the authors´main conclusion, which deserves publication in Plos ONE, thanks!

7. PLOS authors have the option to publish the peer review history of their article (what does this mean?). If published, this will include your full peer review and any attached files.

Reviewer #1: No

---

## [Author Response · Author response to Decision Letter 1]

22 Oct 2020

ACADEMIC EDITOR: While the study has been improved, the interpretation of p62 data needs additional attention. Namely, the unexpected finding that p62 levels were not increased in the presence of pepstatin/E64 could be explained by the relatively short incubation time (6 h). Also, the further increase in p62 levels in the presence of lysosomal proteolysis inhibitors indicates its transcriptional upregulation, which is consistent with its role in delivering autophagic cargo. The authors should briefly address these issues.

We would like to thank the Academic Editor for his comments. We have addressed these points appropriately in lines 533 and 551.

We would further like to thank both the reviewers for their guidance in improving our manuscript content and endorsing it for publication.

---

## [Editor Report · Decision Letter 2]

9 Nov 2020

Enhanced mitophagy in bronchial fibroblasts from severe asthmatic patients

PONE-D-20-12146R2

Dear Dr. Hamid,

We’re pleased to inform you that your manuscript has been judged scientifically suitable for publication and will be formally accepted for publication once it meets all outstanding technical requirements.

Kind regards,

Vladimir Trajkovic

Academic Editor

PLOS ONE
---

## [Editor Report · Acceptance letter]

17 Nov 2020

PONE-D-20-12146R2 

Enhanced mitophagy in bronchial fibroblasts from severe asthmatic patients 

Dear Dr. Hamid:

I'm pleased to inform you that your manuscript has been deemed suitable for publication in PLOS ONE. Congratulations! Your manuscript is now with our production department. 

Kind regards, 

on behalf of

Prof. Vladimir Trajkovic 

Academic Editor

PLOS ONE